## Rapid Communication

hypersalinity; anthropocene; adaptive management; socio-ecological systems; ecological engineering

**Corresponding author:**
Anna R. Armitage;
Email: armitage@tamu.edu

# Global complexities and challenges in the restoration of hypersaline coastal wetlands

Anna R. Armitage[1], Sabine Dittmann[2], Alice R. Jones[3], Jeffrey J. Kelleway[4], Bonani Madikizela[5], Jody O'Connor[6], Francesca Porri[7,8], Kerrylee Rogers[4], Michelle Waycott[9], Christine Whitcraft[10] and Janine B. Adams[11]

[1]Department of Marine Biology, College of Marine Sciences and Maritime Studies, Texas A&M University, Galveston, TX, USA; [2]College of Science & Engineering, Flinders University, Adelaide, SA, Australia; [3]Future Coasts Lab, School of Biological Sciences and Environment Institute, The University of Adelaide, Adelaide, SA, Australia; [4]School of Science, and Environmental Futures Research Centre, University of Wollongong, Wollongong, NSW, Australia; [5]Water Research Commission of South Africa, Pretoria, South Africa; [6]Murray Darling Basin Authority, Adelaide, SA, Australia; [7]National Research Foundation-South African Institute for Aquatic Biodiversity, Rhodes University, Makhanda, South Africa; [8]Department of Ichthyology and Fisheries Science, Rhodes University, Makhanda, South Africa; [9]School of Biological Sciences, University of Adelaide, Adelaide, SA, Australia; [10]Department of Biological Sciences, California State University Long Beach, Long Beach, CA, USA and [11]Institute for Coastal and Marine Research, Department of Botany, Nelson Mandela University, Gqeberha, South Africa

## Abstract

Wetlands in hypersaline environments are especially vulnerable to loss and degradation, as increasing coastal urbanization and climate change rapidly exacerbate freshwater supply stressors. Hypersaline wetlands pose unique management challenges that require innovative restoration perspectives and approaches that consider complex local and regional socioecological dynamics. In part, this challenge stems from multiple co-occurring stressors and anthropogenic alterations, including estuary mouth closure and freshwater diversions at the catchment scale. In this article, we discuss challenges and opportunities in the restoration of hypersaline coastal wetland systems, including management of freshwater inflow, shoreline modification, the occurrence of concurrent or sequential stressors, and the knowledge and values of stakeholders and Indigenous peoples. Areas needing additional research and integration into practice are described, and paths forward in adaptive management are discussed. There is a broad need for actionable research on adaptively managing hypersaline wetlands, where outputs will enhance the sustainability and effectiveness of future restoration efforts. Applying a collaborative approach that integrates best practices across a diversity of socio-ecological settings will have global benefits for the effective management of hypersaline coastal wetlands.

## Impact statement

Restoration of coastal wetlands in the Anthropocene must balance considerations of ecology, economy, and Indigenous rights. These complex and interactive needs require adaptive management in the context of a changing climate, as the effects of sea level rise and shifting precipitation patterns compound with the consequences of land use/land cover change and anthropogenic freshwater demands. Globally, many coastal wetlands are experiencing hypersalinity stress linked to freshwater diversion or drought conditions. These hypersaline wetlands, including those in arid and semi-arid regions, are especially vulnerable to loss and degradation, as increasing coastal urbanization and climate change are rapidly exacerbating freshwater supply stressors. These wetlands present unique management challenges, necessitating the development of novel restoration approaches and success metrics. This article describes restoration successes, challenges, and lessons learned in these habitats, and lays a foundation for developing new, forward-looking restoration strategies that connect the values and needs of human and ecological communities.

## Introduction

Restoration of coastal wetlands in the Anthropocene must account for climate change, where sea-level rise, shifting precipitation patterns and modification of climatic and weather phenomena (e.g., El Niño-Southern Oscillation, cyclones) compound with the consequences of land use/land cover change and anthropogenic freshwater demands. Globally, many coastal wetlands face limited freshwater supply due to drought, flow impoundments by overgrowth of invasive plant species, low precipitation, freshwater diversion and/or groundwater extraction, leading to

hypersaline (exceeding seawater salinity, typically above 40 ppt) conditions (Bornman et al., 2002; Le Maitre et al., 2016; Lovelock et al., 2017; Adame et al., 2021; Duke et al., 2022; Tran et al., 2022). Contemporary definitions of anthropogenic droughts in human-water systems acknowledge the complex interplay of meteorological, geomorphological, hydrological, and anthropogenic drivers (AghaKouchak et al., 2021), where the over-extraction of water can increase the likelihood of drought, irrespective of climatic drivers (Mosley, 2015).

Wetlands in hypersaline settings are typically within coastal estuaries and lagoons that can be intermittently open or closed and may range in vegetation composition and structure from those void of vascular plants (e.g., salt flats or mud flats), to herbaceous or succulent groundcovers, to hypersaline mangrove scrub or short forest. Wetlands in hypersaline environments are especially vulnerable to loss and degradation, as increasing coastal urbanization and climate change rapidly exacerbate freshwater supply stressors (Short et al., 2016; Geedicke et al., 2018), often with critical consequences for foundation species like mangroves or oysters, for ecosystem engineers such as bioturbating organisms (Miller et al., 2017; Lam-Gordillo et al., 2022), or for the conservation of estuarine-dependent fauna (Komoroske et al., 2016; Tweedley et al., 2019; Brookes et al., 2022). Wetlands experiencing acute drought, reduced freshwater inputs, or persistent aridity resulting in hypersalinity pose unique management challenges relative to mesohaline or polyhaline wetlands (with salinity at or below 30 ppt). For example, restoration in hypersaline wetlands may require the use of slower growing, salt tolerant species with lower transplant success rates, potentially delaying ecosystem recovery (Zedler et al., 2003). Thus, hypersaline wetlands require unique restoration perspectives and potentially complex, multifactorial approaches. Given the substantial economic value of the ecological functions of these systems (Davidson et al., 2019), and the cost- and labor-intensive efforts to maintain and restore those functions (Wang et al., 2022), effective outcomes will require consideration of the complex local and regional dynamics that are unique to hypersaline ecosystems. This article considers the challenges facing the restoration and management of these systems, outlines areas needing additional research and integration into practice, and identifies potential paths forward for the future restoration of coastal wetlands subject to hypersalinity.

## Estuarine dynamics

Coastal wetlands occupy a range of geomorphological and climatic settings that influence their form and may periodically create hypersaline conditions. Along high wave energy and/or low precipitation coastlines, intermittent estuaries (also called temporarily closed estuaries) can form in association with sand bars or berms that restrict tidal influence, cutoff low water areas, or perched impoundments (Stein et al., 2021). In some settings, these systems experience low or zero inflow outside of seasonal rainstorms; these low flow and low volume conditions can hover near salinity tolerance thresholds of resident biota. Restoration of these often small, seasonally variable systems is closely linked to watershed inputs, making them highly sensitive to changes in inflow, sediment, nutrients, and other contaminants. Reestablishing dynamic estuary entrances, such as seasonal mouth openings and closures, can improve salinity regimes, enhance intertidal vegetation recovery, and subsequently improve shoreline stability by mitigating erosion,

attenuating waves, and supporting biodiversity (Bilkovic et al., 2016).

Robust baseline data obtained from comprehensive monitoring programs is essential for effective management, especially in low flow and low volume systems (Adams and Van Niekerk, 2020; Stein et al., 2021). A universal challenge is determining appropriate management targets that inform decisions, including management of mouth openings. As in many types of coastal ecosystems, this challenge is difficult because ecological states often shift seasonally (Stein et al., 2021), driven by fluctuations in hydrological, climatic, and marine processes. This seasonality affects water flow, sediment deposition, salinity gradients and species distributions, making it difficult to establish clear reference targets for all expected seasonal states (Little et al., 2017; Mosley et al., 2018).

## Freshwater inflow

Freshwater inflow to coastal wetlands and estuaries is key to maintaining system health and productivity, particularly in arid and semi-arid regions. Rising demand in freshwater abstraction to support growing human populations directly contributes to the salinization and desiccation of coastal wetlands. Scarcity of freshwater can lead to hypersalinization (due to high evaporation; Tweedley et al., 2019) or marinization (extended intrusion of seawater into an estuary; Pasquaud et al., 2012). Additionally, urbanization can lead to reduced seasonal freshwater input while also generating perennial "urban drool," where contaminated freshwater runoff trickles into ephemeral streams during the dry season (White and Greer, 2006; Pilone et al., 2021). Altered freshwater inflow influences estuary mouth states, changes water residence times, and triggers extreme shifts in salinity regimes with consequential biological degradation of mudflats, salt marshes, and mangroves (Zampatti et al., 2010; Dittmann et al., 2015).

Anthropogenic freshwater demands often co-occur with climate change-induced increases in drought frequency and intensity, especially in the wet-dry tropics where coastal estuaries may experience low inflow during the dry season, leading to periodic hypersalinity in the upper intertidal zone. When the wet season is reduced or fails, as can occur with oceanic and climatic perturbations (e.g., El Niño-Southern Oscillation events), the impacts on coastal wetland function can be profound and may cause dieback (including plant mortality in severe instances), especially in mangrove-dominated systems (Duke et al., 2017; Lucas et al., 2017; Otero et al., 2017). In these circumstances, restoration of wetland condition may only be successful when prevailing salinity conditions have returned to a normal state after the perturbation event subsides (Asbridge et al., 2019).

Wetlands in arid systems are already near their tolerance limits in terms of freshwater inputs (Bertness et al., 1992; Howard and Mendelssohn, 1999; Watson and Byrne, 2009; Adame et al., 2021). Therefore, restoring connectivity between freshwater sources and downstream estuaries is key for mitigating the potentially antagonistic effects of anthropogenic freshwater demands and climate drivers, thus enhancing ecological and societal benefits (Arthington et al., 2018b; Adams et al., 2023). However, effective outcomes will require consideration of local and regional dynamics of changing water, sediment, and nutrient inputs from the watershed (Mosley et al., 2023). Adaptive management of hydrological infrastructure may include removing in-stream barriers (e.g., weirs, flood gates) and flood controls on coastal floodplains (e.g., bund walls, levees) to

recreate natural flow and connectivity conditions (Webster, 2010; Chilton et al., 2021). Future restoration efforts will also need to address past overallocation and illegal catchment and abstraction activities. Such management actions must consider future climate projections to ensure restoration is sustainable in a changing socio-ecological framework. In some countries, legal mandates require Environmental Flow (E-Flow) allocation to estuaries and associated wetlands. E-flows describe the volume, timing and duration of flows (the hydrological regime) required to sustain the components, processes and services of estuarine and freshwater ecosystems (Arthington et al., 2018b). These E-Flows safeguard estuarine health and their multiple ecosystem services to society (Arthington et al., 2018a; Adams and Van Niekerk, 2020). Planning and implementation of E-Flow restoration resides with catchment (or watershed) management authorities and should use an adaptive management approach that includes scenario planning, ecological monitoring, and consultation with advisory panels comprised of scientists, stakeholders, and regional Indigenous groups (Rumbelow, 2018). In hypersaline wetlands, however, monitoring, implementation, and enforcement are often underfunded and salinity-specific management is overlooked, especially for invertebrates and other estuarine fauna (Hemeon et al., 2020).

## Landscape modification

Urbanization worldwide has resulted in substantial structural and physical modifications of shorelines and watersheds in general and for intermittently closed estuaries in particular (Bugnot et al., 2021; Lawrence et al., 2021). Resulting changes to erosion, freshwater inputs, and deposition patterns disrupt coastal wetland hydrodynamics (Dugan et al., 2018), potentially altering salinity regimes in systems near biotic salinity tolerance limits (Whitfield et al., 2012). Construction of structures intended to manage erosion (e.g., seawalls, breakwaters), can fragment wetlands and restrict water flow (Bulleri and Chapman, 2010). Further, upland development may lead to the loss of relict coastal wetlands due to coastal squeeze, further compromising ecological functionality (Munsch et al., 2017) and reducing biodiversity (Bulleri and Chapman, 2010; Dugan et al., 2018). Coastal wetland restoration in heavily regulated, urbanized systems with competing water demands (Verdonschot et al., 2013), such as those in arid and semi-arid regions, present unique challenges. While full recovery to 'pristine' pre-disturbed states is often unachievable, adaptive eco-engineering approaches (both hydrological and ecological remediation) may help retain the remaining ecosystem values of coastal wetlands (Elliott et al., 2016; Zedler, 2017).

## Multiple co-occurring stressors

Hypersaline coastal wetlands and estuaries face multiple, cumulative long-term stressors that can complicate restoration and management planning. For example, the impacts of drought and high salinity conditions often coincide with other climate-driven stressors including fire (Taillie et al., 2019) and freeze events (Madrid et al., 2014; Osland et al., 2017). Likewise, erosion or sedimentation following severe storms and floods might be amplified during post-drought periods when vegetation cover is reduced, often slowing ecosystem recovery (Cahoon, 2006; Alexandra and Finlayson, 2020). Drought or hypersalinity may intensify the consequences of anthropogenic stressors associated with land-use type and intensity, such as surface or groundwater extraction, nutrient input, and

agricultural grazing (e.g., Tran et al., 2019). Broadly, interactions between hypersalinity and other stressors often constrain ecosystem productivity and restoration potential (Box 1). In many cases, specific outcomes of interactive stressors are specific to sites, species, and stressor conditions, and predicting these patterns will require ongoing and new research efforts (Morzaria-Luna et al., 2014).

Any restoration activities in these systems will need to consider the complex range of acute and chronic stressors that may be concurrently or sequentially affecting an ecosystem (Turner II et al., 1990; Kondolf and Podolak, 2014; Spencer and Lane, 2016). Furthermore, what works well for a foundational species in one region may not transfer to other portions of its range (Box 1). Managing multiple and compounding stressors is especially challenging given projections of increasing frequency and intensity of multiple co-occurring climatic stressors (He and Silliman, 2019), and a lack of understanding and difficulty predicting the synergistic interactions of co-occurring stressors (Stockbridge et al., 2024).

## Values of local and Indigenous peoples

The recognition and appreciation of Traditional and Local Knowledges are on the rise, and along with stakeholder values, they are now considered critical for enhancing coastal ecosystem restoration and management success (e.g., Uprety et al., 2015; Hemmerling et al., 2019; Loch and Riechers, 2021), including wetlands (de Oliveira et al., 2024). Despite the recognized value of Indigenous and Local Knowledges and efforts to rectify skewed western epistemologies (Parsons and Fisher, 2020) and inequities through international commitments (e.g., UN Declaration on the Rights of Indigenous People, Kunming-Montreal Global Biodiversity Framework, and others), the active participation of Indigenous communities in wetland ecosystem restoration remains under-utilized (Gaspers et al., 2022; Reed et al., 2022). Real collaborations between wetland custodians and conventional knowledge scientists, policy makers and practitioners (Muller, 2012; Parsons and Fisher, 2020) are still limited. Without input from people that reside in and sustainably use the resources within coastal systems, restoration and management actions risk degrading ecosystems and further loss of critical ecosystem services (Peer et al., 2022; Nsikani et al., 2023). This threat is particularly potent in arid, hypersaline wetland systems nearing the biotic tolerance limits for salinity, where "standard" restoration approaches, such as managed realignment, re-establishment of water flow, sediment and nutrient control, and revegetation (Almendinger, 1998; Henry et al., 2024) are less likely to be effective. Thus, emphasizing the integration of Indigenous, traditional, and locally-led community knowledge in wetlands research, management, and governance is crucial in these hypersaline habitats, offering tangible environmental benefits by informing ecologically sustainable (nature-based) approaches (Seddon et al., 2021; Reed et al., 2022) that are collectively relevant (Pyke et al., 2018). For example, Indigenous-led workshops can be part of a decentralized framework that supports community (including youth and elderly) leadership and rights of custodians to promote meaningful review of needs, co-design and co-implementation of restoration/management (Gann et al., 2019; Dickson-Hoyle et al., 2021; Robinson et al., 2021), governance (de Oliveira et al., 2024) and ecosystem stewardship (Holmes and Jampijinpa, 2013) of arid wetlands.

Box 1 Case study: Multiple co-occurring stressors in hypersaline mangrove wetlands.

Multiple stressors decrease ecosystem productivity. Black mangroves, *Avicennia germinans,* occur across the Gulf of Mexico (North America) coastline, but face unique combinations of multiple stressors and vulnerabilities in different ecoregions. At their range edge in the northern Gulf of Mexico, *A. germinans* adapts to hypersaline conditions and freeze risk by developing dense complexes of narrow xylem vessels; this "safe hydraulic architecture" reduces the risk of embolism and cavitation in high salinity and/or freezing conditions (Madrid et al. 2014). However, that adaptation also reduces absolute conductance capacity and diminishes structural integrity, so *A. germinans* in this region are shorter (left image) and more prone to fracture. In contrast, *A. germinans* in warmer, wetter ecoregions, such as south Florida (right image), have more vessel area and are thus taller and more productive.

Multiple stressors present unique management challenges. Restoration or management actions that are effective in south Florida (right) are unlikely to transfer to hypersaline regions with multiple abiotic stressors (left), even though the same mangrove species is dominant in both systems.

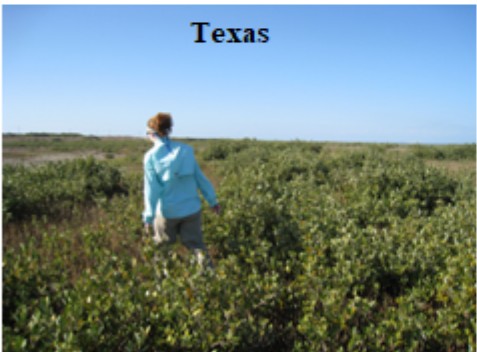

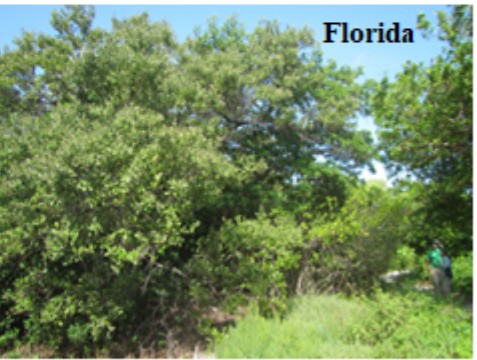

Texas: Scrub-shrub growth form of *A. germinans* in a marginal habitat with multiple concurrent abiotic stressors. Center: Mangrove scientist for scale.

Florida: Tall *A. germinans* canopy in an ideal habitat with relatively few abiotic stressors; note mangrove scientists for scale in the lower right.

## Future restoration in practice

Coastal ecosystem restoration demands an integrated, adaptive, and often long-term approach that recognizes changing climatic conditions and increasing anthropogenic pressures. To develop holistic restoration strategies within the Anthropocene context, the following considerations are suggested as critical for the management of hypersaline wetlands:

### Socio-ecological framework

Adopting a socio-ecological systems framework is crucial, incorporating all stakeholders and balancing societal and ecological benefits (Adams et al., 2020; Nsikani et al., 2023). This framework should embrace transdisciplinary approaches that explicitly integrate Indigenous and Local Knowledges, promote Indigenous-led restoration, and engage local communities in restoration practice. Collaborative partnerships among community stakeholders and regulatory agencies are essential for co-producing design and management strategies in hypersaline wetlands. These partnerships will foster sustainable relationships and ensure long-term provision of essential ecosystem functions and the unique suite of biota that are adapted to these hypersaline systems.

### Ecological engineering

Opportunities for "Engineering with Nature" designs (Bridges et al., 2018), hold promise for restoring hypersaline wetland systems, especially along heavily modified shorelines (Elliott et al., 2016). Diverse approaches (e.g., managing upstream and downstream infrastructure, constructing novel habitat, and reintroducing foundation species such as salt-tolerant mangroves) can lead to some measure of restoration success. Decisions to pursue engineered solutions should be carefully balanced against the benefits and risks of passive approaches that allow for ecosystem restoration to follow an unmanaged trajectory. In some instances, active restoration work can be ecologically successful and a publicity boon (e.g., Banerjee et al., 2023), but can also sometimes yield incremental

ecological outcomes (e.g., Lee et al., 2019). Engineered solutions may not be responsive or adaptable to rapidly changing climate conditions, including increased frequency and intensity of extreme events (Ting et al., 2019; Cohen et al., 2021), or to chronic and irreversible stressors such as sea level rise (Saintilan et al., 2022). Given the uncertainty and variability facing hypersaline wetland systems, and the lack of baseline data to inform management targets, it may be challenging to develop sustainable, long-lived engineered designs that can adaptively respond to future climatic conditions.

### Regulatory framework

In complex hypersaline systems that extend across socio-political borders, policy provisions to guide the prioritization and management of water allocations for environmental purposes (E-flows) are being incorporated into some legal agreements for hypersaline systems such as Australia's Murray Darling Basin Plan (MDBA, 2012) and the Colorado River Minute 323 (IBWC, 2017). In some cases, legally mandated E-flow requirements have bolstered water security by increasing flows, thus generating drought protection to end-of-catchment coastal wetlands (Brookes et al., 2023). In many other instances, however, there remains substantial room for cross-agency collaboration and monitoring to improve data-informed guidance for inflow and freshwater allocation decisions at the catchment scale (Davis et al., 2015).

### Adaptive management

Future restoration of hypersaline systems must integrate climate change projections and anticipated impacts on wetlands and associated communities. For example, managers should consider the delivery of freshwater flows and restoration efforts in the context of drier futures with expanding human populations and subsequent demands on upstream water resources. Addressing these challenges will involve difficult decisions about human-environmental trade-offs that consider the salinity setting (Largier, 2023) and the local socio-ecological framework as described above. In doing so, restoration practitioners may need to prepare people for alternate environmental, social and economic futures while striving to restore to the 'best possible' states under a changing climate.

Climate change poses adaptive management implementation challenges in hypersaline systems, as this has shifted climatic and rainfall baselines and increased unpredictability in rainfall and extreme events, impacting freshwater use and delivery to estuaries (Stein et al., 2021). Such impacts are likely to also affect sediment supply to coastal wetlands, which is already low in most arid/semi-arid areas. Any further reduction in sediment supply due to reduced freshwater/land-based inputs to the coast will subsequently reduce accretion rates in wetlands. This will decrease the ability of these systems to maintain their optimal position in the tidal frame and lead to increased erosion and/or shoreline submergence with sea-level rise. These climate-induced changes may affect the state of estuaries post-restoration, necessitating revised management practices, notably a "learning-by-doing" approach.

### Next steps

Restoration is vital to maintain and improve the health of hypersaline wetlands, ensuring the provision of multiple ecosystem services to society. There are unique challenges associated with adaptive restoration of wetlands subject to salinity extremes, and these

challenges are compounded by co-occurring stressors and anthropogenic alterations, including estuary mouth closure and freshwater inflow diversions. Restoration in practice should be adaptively informed by locally-led, community-informed best practices at the catchment scale, and future research should seek to fill gaps in this type of knowledge. There is a broad need for actionable research on adaptively managing high-salinity wetlands that will enhance the sustainability and effectiveness of future restoration efforts. Using practices, information, and lessons shared across a diversity of socio-ecological settings will improve the effective management of hypersaline coastal wetlands on a global scale.

**Open peer review.** To view the open peer review materials for this article, please visit http://doi.org/10.1017/cft.2025.1.

**Data availability statement.** No new data are reported in this article.

**Acknowledgements.** This article was inspired by discussions during and after an organized special session titled "Adaptive habitat management in a changing climate: challenges in the ecological and cultural restoration of coastal wetlands in regions vulnerable to drought conditions" at the Society for Ecological Restoration 10th World Conference on Ecological Restoration in Darwin, Australia in September 2023. The meeting was held on country of the Larrakia Nation. We acknowledge the Indigenous custodianship of the Larrakia saltwater people, and the coastal wetland custodianship of Indigenous people globally.

**Author contribution.** A.R.A., J.B.A., C.W., and K.R. conceived the paper concept and organized a special session at the Society for Ecological Restoration 10th World Conference on Ecological Restoration that was attended by most authors. A.R.A. led the writing and figure design. All authors contributed to writing and editing the text.

**Financial support.** The National Research Foundation of South Africa through the support of the DSI/NRF Research Chair in Shallow Water Ecosystems supported J.B.A. (UID 84375). Funds were provided to F.P. by the National Research Foundation of South Africa (Grant Number 136486; Reference: MCR210218586984). Travel for K.R. was supported by the Australian Research Council (DP210100739).

**Competing interest.** The authors declare none.

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
