## [Reviewer Report]

There is much to like in this paper. It is well written and well argued. It is highly appropriate for the journal and covers an environmnet - hypersaline wetlands - that is not well represented in the literarture. I particularly like the sections which drew attention to the need to pay attention to the interaction of ‘upstream’ and ‘downstream’ effects, an attention sadly lacking in much of our siloed literature. There are some structural issues - what text goes where - and there are a few places where unsupported statements do need some back up with appropriate referencing. But these issues should be readily fixable. But... having said all this, I do feel that the later stages of the paper rather lose the Hypersaline environments focus and, becoming more polemical, drift into more general etxt on coastal wetalnd restoration. I began to move from ‘minor revision’ to ‘major revision. as a consequence. That’s a shame because there really is a much, much better paper embedded in the current text. Finally, I don’t think Box 1 adds much to the argument. I would remove. What would be really nice would be a single figure cartoon contrasting ’bad‘ hypersaline settings / management. ’good' settings / management. That might get widely picked up and the paper widely quoted as a result. Not essential but, in my view, well worth thinking about.

---

## [Reviewer Report]

I have now had the chance to read closely the revised manuscript and to consider the authors' very thorough and very clear responses to the issues raised on the original submission. In each and every case the authors have taken on board the criticisms raised and revised the manuscript accordingly. I particularly appreciate the complete re-write of section VII which has greatly improved the focus of the manuscript. Oh that all authors would respond so fully and so positively; a great example of how the review and revision process should work in academic journals. Very impressive. I still feel that Box 1 should be omitted but I am OK with the authors and the reviewer continuing to disagree on this point. I note also that the authors have added both an impact statement and a graphical abstract. All in all this is a very strong response. My view is that the manuscript should now be accepted for publication.

---

## [Editor Report]

The authors had responded to the comments in accordance with comments from the reviewer. Since the reviewer had gone through all responses made in the revision, and thought it satisfied to the comments and suggestion raised, I also agreed with the recommendation of acceptance suggested by the reviewer.